# Unravelling the Epigenome of Myelodysplastic Syndrome: Diagnosis, Prognosis, and Response to Therapy

**DOI:** 10.3390/cancers12113128

**Published:** 2020-10-26

**Authors:** Danielle R. Bond, Heather J. Lee, Anoop K. Enjeti

**Affiliations:** 1Faculty of Health and Medicine, School of Biomedical Sciences and Pharmacy, University of Newcastle, Callaghan, NSW 2308, Australia; Heather.Lee@newcastle.edu.au; 2Department of Haematology, Calvary Mater Newcastle, Waratah, NSW 2298, Australia; Anoop.Enjeti@calvarymater.org.au; 3Faculty of Health and Medicine, School of Medicine and Public Health, University of Newcastle, Callaghan, NSW 2308, Australia; 4Department of Haematology, NSW Health Pathology–Hunter, New Lambton Heights, NSW 2305, Australia

**Keywords:** myelodysplastic syndrome, DNA methylation, long non-coding RNA, micro-RNA, diagnosis, prognosis, treatment

## Abstract

**Simple Summary:**

Myelodysplastic syndrome (MDS) is a type of blood cancer that mostly affects older individuals. Invasive tests to obtain bone samples are used to diagnose MDS and many patients do not respond to therapy or stop responding to therapy in the short-term. Less invasive tests to help diagnose, prognosticate, and predict response of patients is a felt need. Factors that influence gene expression without changing the DNA sequence (epigenetic modifiers) such as DNA methylation, micro-RNAs and long-coding RNAs play an important role in MDS, are potential biomarkers and may also serve as targets for therapy.

**Abstract:**

Myelodysplastic syndrome (MDS) is a malignancy that disrupts normal blood cell production and commonly affects our ageing population. MDS patients are diagnosed using an invasive bone marrow biopsy and high-risk MDS patients are treated with hypomethylating agents (HMAs) such as decitabine and azacytidine. However, these therapies are only effective in 50% of patients, and many develop resistance to therapy, often resulting in bone marrow failure or leukemic transformation. Therefore, there is a strong need for less invasive, diagnostic tests for MDS, novel markers that can predict response to therapy and/or patient prognosis to aid treatment stratification, as well as new and effective therapeutics to enhance patient quality of life and survival. Epigenetic modifiers such as DNA methylation, long non-coding RNAs (lncRNAs) and micro-RNAs (miRNAs) are perturbed in MDS blasts and the bone marrow micro-environment, influencing disease progression and response to therapy. This review focusses on the potential utility of epigenetic modifiers in aiding diagnosis, prognosis, and predicting treatment response in MDS, and touches on the need for extensive and collaborative research using single-cell technologies and multi-omics to test the clinical utility of epigenetic markers for MDS patients in the future.

## 1. Introduction

Myelodysplastic syndrome (MDS) is a malignant disease characterised by inefficient haematopoiesis and cytopenias [1]. It commonly affects the ageing population (>65 yrs) and is predicted to rise in incidence. There is a high economic burden associated with MDS due to high costs of chemotherapy, infections and supportive care [2]. Bone marrow transplantation is the only curative option for MDS [3]. However, most patients are unfit for transplantation. Those that receive chemotherapy often either don’t respond or stop responding to therapy and progress to bone marrow failure or acute myeloid leukemia (AML). There are currently no new clinical tests or markers for diagnosis, prognosis or predicting treatment response in MDS.

## 2. MDS Diagnosis and Prognosis

MDS is initiated when a hematopoietic stem cell acquires mutations leading to uncontrolled proliferation of dysplastic blasts in the bone marrow (BM) and peripheral blood (PB) (<20%) [4]. These dysplastic myeloid blasts are non-functional and incapable of differentiating, resulting in cytopenia and ineffective haematopoiesis [5]. MDS is routinely diagnosed when full blood count abnormalities are followed up with bone marrow biopsy. The latter is used for cytogenetic, genetic, histochemical, and flow cytometric analysis to examine chromosomal, genetic and morphological abnormalities for diagnosis of MDS [4]. The WHO classification system is based on peripheral blood and bone marrow morphological features including presence of dysplasia and percentage of blasts [6]. Revised International Prognostic Scoring System (IPSS-R), which groups patients into risk categories based on the percentage of immature blood cells in the bone marrow, complete blood count results and the pattern of cytogenetic abnormalities provides a clinical assessment risk score [7]. This scoring system aids in clinical decision, prognosis, and risk of progressing to AML.

## 3. Treatment Options for MDS

The treatment for low-risk MDS consists of supportive care such as blood and platelet transfusions, antibiotics, erythropoietin (EPO) injections to stimulate red blood cell production and granulocyte colony-stimulating factor (G-CSF) to stimulate white blood cell production [8,9]. Hypomethylating agents (HMAs) are used in higher risk patients to achieve remission, improve blood counts, reduce transfusion dependence, and bridge suitable patients to allogenic transplantation. HMAs such as azacytidine (AZA; Vidaza) [10] or decitabine (DAC; Dacogen) are used in frontline therapy [11]. While HMAs are effective in around 50% of MDS patients in the short term, many patients become resistant to therapy and progress to bone marrow failure or to AML [12]. New targeted therapeutics such as BCL2 inhibitors (Venetoclax) [13,14,15] and immune checkpoint inhibitors [16,17,18] are currently being tested in combination with HMAs or as a monotherapy following HMA failure. In addition, lenalidomide (Revlimid) is FDA approved for use in MDS with 5q deletion and immunosuppressive therapy or other chemotherapies (e.g., cytarabine) may be used as needed to improve normal blood cell production and reduce blast counts before stem cell transplantation, respectively [19]. The only curative treatment for MDS is stem cell transplantation. However, most patients are too frail or have existing comorbidities which precludes them from undergoing this curative option [8,10].

## 4. MDS Pathophysiology

MDS, similar to other cancers, is thought to be initiated by the accumulation of mutations (driver mutations) that lead to positive selection and clonal outgrowth of the malignant clone [20,21]. There are many other factors that contribute to this process such as epigenetic changes [22], the bone marrow micro-environment [23,24] and extrinsic factors (autoimmunity, previous chemoradiotherapy) [20]. Research over the last decade involving next-generation sequencing technology, has been able to detect driver alterations in MDS related to chromosomal and copy number abnormalities, and somatic mutations. Chromosomal abnormalities are found in ~50% of MDS patients, with the most frequent being -7/del(7q) and -5/del(5q) [25,26]. These changes can also co-occur as complex karyotypes and are often accompanied by *TP53* mutations [20,27,28,29]. Interestingly, a recent study has highlighted an important role for bi-allelic *TP53* mutations [29]. There was no difference in outcome for patients with wild-type or mono-allelic mutations of *TP53*. However, multiple mutations in *TP53* were able to predict outcomes independent of the revised international prognostic scoring system.

MDS patients carry a median of 9 somatic mutations within the exome, this includes both driver and passenger mutations, which is considerably less than most solid cancers [30]. More than 30 driver mutations have been identified in MDS, typically patients harbour 2 or 3 driver mutations, the number increasing with risk severity [20,31,32]. These driver genes can be categorised into distinct functional pathways involving DNA methylation, RNA splicing, chromatin modification, transcription, signal transduction and others. Some of the most frequently mutated genes in MDS belong to pathways such as RNA splicing (*SF3B1, SRSF2, U2AF1, U2AF2, ZXRSR2, SF1,* and *SF3A1*) or epigenetic regulation [20]. The latter being involved in DNA methylation (*DNMT3A, TET2, IDH1/IDH2*) or chromatin/histone modification (*MLL2, EZH2,* and other PRC2 components, *ARID2* and *ASXL1*) [20]. Therefore, this highlights the importance of epigenetics such as changes to DNA methylation and histone modifications in the pathogenesis of MDS.

## 5. Epigenetic Modifiers

Cancer is typically defined by the accumulation of genetic mutations that lead to uncontrolled cell division. However, other factors such as epigenetics are known to also play a pivotal role in cancer initiation and progression [33]. Epigenetics which translates as the study of factors “on top of” (epi) genes, describes mechanisms that can modify gene expression without changing the DNA sequence itself [34]. Therefore, epigenetic factors act as a master switch, having the capability to regulate gene expression. While genetic modifications consist of mutations in tumor suppressor genes and oncogenes, epigenetic modifications are typically more complex and comprise changes in DNA methylation, chromatin structure, histone modifications, nucleosome remodelling, and non-coding RNAs [33]. During the development and progression of MDS, a myriad of epigenetic changes has the propensity to affect gene expression and cellular function, many of which have untapped potential in aiding clinical decision making throughout the course of a patient’s journey with MDS.

### 5.1. DNA Methylation

DNA methylation is the addition of a methyl group (-CH3) to the 5′ carbon of cytosines that are followed by a guanine (CpG sites), which results in 5-methylcytosine (5mC) (Figure 1A) [35]. This reaction is catalysed by a family of enzymes known as DNA methyltransferases (DNMTs), and include DNMT1, DNMT3A and DNMT3B [35]. DNMT3 isoforms are responsible for adding new methylation marks to DNA (de novo methylation) at loci which were previously unmethylated, whereas DNMT1 is known primarily as the maintenance enzyme, since it is responsible for maintaining methylation marks on the newly-synthesised strand after DNA replication (Figure 1B) [35,36]. However, all three function together to maintain methylation marks during DNA replication, particularly in CpG-dense regions [37]. The removal of methylation marks is initiated by ten-eleven translocase (TET) enzymes, namely TET1 and TET2, by oxidising 5mC to 5-hydroxymethylcytosine (5hmC), which can then undergo base-excision repair (BER), converting back to an unmodified cytosine (Figure 1) [35]. Methylation predominately occurs at CpG poor regions and at repetitive elements, whereas CpG dense regions (termed CpG islands) are usually lacking methylation in normal somatic cells [35,38].

DNA methylation in gene promoters influences transcription factor binding and chromatin structure [39,40] leading to transcriptional repression as methylation blocks interactions between transcription factors and the DNA, or facilitates binding of repressive factors, resulting in decreased gene expression [41,42]. In contrast, DNA methylation in gene bodies influences transcriptional activation [40] and RNA splicing [43,44], leading to increased gene expression. Therefore, changes in DNA methylation can impact a multitude of genes and thus cellular functions. It is not surprising that mutations in DNMTs and TETs are observed in cancers, particularly MDS and AML. These mutations in DNA methylation machinery are known to influence global DNA methylation changes observed in cancers, e.g., *DNMT3A* mutations in AML are associated with genome-wide hypomethylation [45,46]. Most solid malignancies display global hypomethylation with hypermethylation present at specific sites in the genome [47]. Interestingly, MDS is typically characterised by global hypermethylation, and this may explain why MDS patients respond well to HMAs [48].

### 5.2. Non-Coding RNAs

Up until a couple of decades ago, 98% of the genome within each cell was considered “junk” DNA due to its non-coding nature, i.e., does not code for any proteins [49,50,51]. Since then, it was discovered that these areas of the genome harbour non-coding RNAs (ncRNAs) that act like a switch to turn genes on or off, hence regulating gene expression. There are different classes of non-coding RNAs typically grouped by size, with small ncRNAs such as micro-RNAs (miRNAs) and piwi-interacting RNAs (piRNAs), and larger ncRNAs such long non-coding RNAs (lncRNAs) [51]. miRNAs and lncRNAs in particular, have been shown to play functional roles in diseases such as cancer [51].

#### 5.2.1. Micro-RNAs

miRNAs are small, ncRNAs (~22 nucleotides) that are found in plants and animals [51]. They contain a “seed” region (~6–8 nucleotides) that binds to the 3′UTR of target mRNA transcripts via sequence complementarity, resulting in mRNA decay or inhibition of translation [52,53]. Therefore, miRNA function in post-transcriptional gene regulation, which results in decreased protein expression of target mRNA. Each miRNA can potentially target hundreds of mRNAs, some of which may belong to the same pathways or pathways with similar functions [52]. Many miRNAs have been shown to play a pivotal role in cancer and cancer progression, in which changes to the expression of specific miRNAs have led to the disruption of key pathways or proteins that are important in cancer biology [54]. For example, miRNAs that target tumour suppressor genes are typically upregulated in cancers, as this prevents expression of tumour suppressors and supports the growth of cancers [53,55]. Conversely, many miRNAs that target oncogenes are commonly downregulated to allow the expression of oncoproteins that drive cancer initiation and progression [53,55]. The key miRNAs which have been described to have a role in the pathophysiology of MDS are discussed below.

#### 5.2.2. Long Non-Coding RNAs

LncRNAs are long, non-coding transcripts (>200 nucleotides) that do not encode proteins [51]. There is potentially more than 15,000 lncRNAs expressed in the human genome, and they have been shown to function in many ways [56]. LncRNAs can recruit different components of the chromatin remodelling complex to change chromatin organisation [57,58]. They can act as a sponge by binding to miRNA via base complementarity and therefore reduce the effects of miRNA, and they can enhance or inhibit transcription [57,58]. LncRNAs can affect cellular functions via a range of mechanisms, and it is no surprise that these molecules are exploited in different types of cancers. They have been shown to modulate cancer cell proliferation, migration, immune escape and apoptosis, among other common features of cancer progression [51,59]. For example, a lncRNA that acts as a sponge for an anti-tumour miRNA (targets oncogenes) would result in upregulated expression of oncogenes which promotes tumour initiation and/or progression. Indeed, this has been shown recently in gastric cancer with the lncRNA UCA1 [60].

## 6. Epigenetic Modifiers That Aid in the Diagnosis of MDS

Given the importance of DNA methylation and ncRNAs in cancer biology, epigenetic modifiers in MDS, including changes in DNA methylation, miRNAs and lncRNAs, and how they may aid in MDS diagnosis, prognosis and predicting response to treatment will be discussed below.

### 6.1. DNA Methylation as a Diagnostic Tool for MDS

Diagnostic testing is usually initiated once patients have become symptomatic and cytopenias are prominent. Some of the most mutated genes in MDS are members of the DNA methylation machinery such as *DNMT3A, TET2, IDH1* and *IDH2* [61]. Mutations in *DNMT3A* and *TET2* have been observed in clonal haematopoiesis and early in MDS [62,63]. These mutations often lead to global changes in DNA methylation or pronounced changes at specific genomic sites. Mild cytopenias without overt features of myelodysplasia within the bone marrow are now increasingly recognised such as clonal cytopenias of uncertain significance (CCUS) [64]. Whether DNA methylation signatures may have the potential to aid in the recognition of pre-MDS states such as CCUS or Clonal Haematopoiesis of Indeterminate Potential (CHIP) needs to be determined by prospective studies [64].

Analysis of 5mC in bone marrow mononuclear cells from MDS patients using immunocytochemistry showed that ~85% of cases displayed significantly higher levels of 5mC compared to control patients with anaemia of chronic disease [65]. This suggests that in MDS, detection of 5mC levels which are indicative of hypermethylation, may be a useful tool in diagnosing MDS. Indeed, DNA hypermethylation (especially hypermethylation at enhancers) is commonly observed in MDS, particularly in cases involving *TET2* loss of function mutations [66,67].

DNA methylation changes at specific sites in the genome have also been observed in MDS (Table 1). It was recently shown that CpG island methylation associated with six genes (*ABAT, DAPP1, FADD, LRRFIP1, PLBD1*, and *SMPD3*) in bone marrow cells is a marker of MDS, and could diagnose MDS with 95% specificity and 91% sensitivity [68]. Another group has also shown significantly increased *ABAT* methylation and decreased ABAT gene expression in MDS compared to controls [69]. Significantly higher gene-specific promoter methylation of *SOX7* (55% of patients) [70], *ID4* [71], *SOX17* [72], *DLX4* [73], *GPX3* [74], *DLC-1* [75], *CDKN2A/B* [76], and WNT antagonists (*sFRP1/2/4/5, DKK-1/3*) [77] have also been found in MDS. Moreover, significantly higher *ID4* gene promoter methylation could distinguish MDS from aplastic anaemia, which can be challenging particularly MDS with a low blast count, hypoplasia and/or normal karyotype [71]. Hypomethylation of the *let-7a-3* promoter has also been observed in MDS patients compared to controls [78]. Overall, global DNA methylation levels and methylation at specific sites show promise as biomarkers for the diagnosis of MDS. However, for DNA methylation markers to be utilised in MDS diagnosis, they would need to be validated in patient cohorts and ideally in peripheral blood mononuclear cells. The latter would provide a less invasive test to diagnose MDS using peripheral blood markers without the need for frequent, invasive bone marrow aspirates.

### 6.2. miRNA and lncRNA Signatures for the Diagnosis of MDS

The expression levels of ncRNAs such as miRNAs and lncRNAs are dysregulated in MDS, and therefore may also aid in diagnosis (Table 1). Many of the genes listed in the above-mentioned 6-gene methylation signature are targets of miRNA and lncRNA with expression changes in MDS. This study found 72 miRNAs and 214 lncRNAs with significant differential expression in MDS together with gene expression and methylation changes compared to healthy controls, forming an integrative network that may aid in the diagnosis of MDS [103]. In addition, overexpression of the DLK1-DIO3 region, which harbours a large miRNA cluster and *MEG3* (lncRNA) gene promoter, was observed in 50% of patients before treatment with AZA, and this was in conjunction with the diagnosis of AML with myelodysplasia-related changes [79]. Therefore, overexpression of the miRNA cluster before treatment may aid in the diagnosis of AML with myelodysplasia-related changes in higher risk MDS patients. Another group found a co-expression signature which contained 6 differentially expressed lncRNAs that were co-expressed with ABAT in MDS patients [96]. The expression of one of these lncRNAs (lncENST00000444102) and ABAT were significantly downregulated in MDS [96].

#### 6.2.1. miRNAs

Studies over the last decade have started to provide evidence for the potential clinical utility of miRNA expression profiling in the diagnosis of MDS (Table 1). Early studies found miRNA signatures that discriminated MDS from healthy controls, such as miR-378 [80], miR-632 [80], miR-636 [80] and let-7 family members [81]. miRNA expression profiling has also been able to discriminate between risk groups [81,84] and between MDS with chromosomal alterations and normal karyotype [104]. A higher percentage of miRNAs has also been observed in low-risk MDS, compared to controls and high-grade MDS [82]. More recently, increased expression of haematopoiesis-related miRNAs (miR-34a, miR-125a and miR-150) were observed in MDS, and higher expression of miRNAs clustered on 14q32 was found in early MDS [83].

The following miRNAs have shown increased expression in MDS: miR-17-92 cluster [84], miR-222 and miR-10a [81], miR-194-5p (AUC 0.797) and miR-320a (AUC 0.729) [85], miR-21 [86,87], miR-34b [88], miR-661 [89], miR-720 [87] and miR-205-5p (AUC 0.825) [90]. Conversely, downregulation of the following miRNAs has been observed in MDS: miR-124 [91], miR-155, miR-182, miR-124a, miR-200c, miR-342-5p and let-7a [93], miR-146a, miR-150 and let-7e [81], miR-143 [92], miR-671-5p and miR-BART13 [87]. In some cases, these changes in miRNA expression have been correlated to changes in DNA methylation in their promoters. Increased expression of miR-34b is associated with hypomethylation [88], and decreased expression of miR-124 is linked to increased DNA methylation [91]. Some of these miRNAs are strongly associated with MDS (AUC close to 1) and can accurately distinguish MDS from healthy controls. However, only miR-21 [86,87], miR-150 [81,83] and let-7 miRNAs [81,93] have been found to be differentially expressed in MDS by more than one research group.

Another area of interest involving miRNAs, is their presence in extracellular vesicles (EVs) in the plasma of MDS patients. EVs, such as exosomes, contain cargo that consists of small RNAs and miRNAs that can be delivered to cells via intercellular communication [105]. Two recent studies have explored the expression of miRNAs in EVs or exosomes in MDS patients. Enjeti et al. 2019 [94] observed significantly higher numbers of small RNAs and miRNAs in EVs from plasma of red-cell transfusion-dependent MDS patients, with upregulated expression of miR-548j and miR-4485, and down-regulation of miR-28 and let-7d. Another group found 21 exosomal miRNAs with strong association with MDS [95]. They also found 7 miRNAs that were present in both MDS and severe aplastic anaemia with strong association such as miR-378i (AUC 0.99), miR-574-3p (AUC 0.87), miR-196a-5p (AUC 0.85), miR-3200-3p (AUC 0.83) and miR-196b-5p (AUC 0.79) [95]. Therefore, although not routinely utilised in the clinic yet, exosomal miRNAs may prove to be a useful tool in the diagnosis of MDS.

#### 6.2.2. lncRNAs

Hypermethylation of the *MEG3* gene promoter was observed in 35% of MDS cases in 2010, which was the first study implicating a lncRNA in MDS [106]. Since then, more studies have analysed the expression of specific lncRNAs and there has also been a shift towards exploring the global profile of lncRNA expression in MDS (Table 1). Knowledge of global changes in lncRNAs is important to better understand how they are globally influencing cancer cell functions given their complexity in mode of action and potential to interact with multiple targets. A study in 2013 had an interesting finding, in which conditional deletion of the lncRNA XIST in hematopoietic cells of mice, which is required for X chromosome inactivation during embryogenesis, led to a highly aggressive mixed MDS and MPN phenotype with complete penetrance [107]. This suggests that the lncRNA, XIST, protects hematopoietic cells from malignancy.

Since these early studies, more lncRNAs have been found to display deregulated expression in MDS using global profiling of CD34+ BM cells from MDS patients and these include: linc-BDH1-1, linc-FAM75A7-7, linc-HHLA2-2, linc-JMJD1C-3, linc-PRKD1-2 and linc-RPIA [98], as well as TC07000551.hg.1, TC08000489.hg.1, TC02004770.hg.1, and TC03000701 [99]. Overexpression of CCAT2 was also observed in MDS patient CD34+ BM cells and mononuclear PB cells compared to healthy age matched controls [100]. In addition, increased expression of a novel lncRNA, LOC101928834, was found in MDS and AML, and could discriminate MDS-RAEB patients from controls (AUC 0.9048) [101]. Significantly decreased expression of LEF1-AS1 has also been shown in MDS compared to healthy controls [97]. Lastly, with the recent advent of single-cell technologies, gene expression profiling of lncRNAs in single cells from MDS patients (CD34+ aneuploid cells) has started to highlight deregulated lncRNAs and the pathways they are involved in. This study found 590 downregulated lncRNAs which are involved in immune response, cellular response and gene expression, and DNA damage response [102]. Conversely, the 372 upregulated lncRNAs were associated with cell metabolism and cell signalling [102]. Our understanding of the functional roles of lncRNAs and their utility as diagnostic biomarkers in MDS is still yet to be thoroughly tested and confirmed.

## 7. Epigenetic Modifiers That Are Associated with MDS Prognosis

### 7.1. DNA Methylation Signatures That Predict Prognosis

DNA methylation changes have also been associated with predicting prognosis in MDS patients of various sub-groups, particularly with regards to overall survival (OS) (Table 2). High methylation levels globally across the genome have been associated with significantly lower OS and increased progression to AML. However, on multivariate analysis it was not an independent variable for OS or progression [108,109]. A recent publication grouping MDS patients into DNA methylation clusters has identified subtypes that are genetically distinct and correlate with OS [110]. In addition, hypomethylation of *CD93* in MDS patients resulted in shorter OS rates [110] and MDS patients with *let-7a-3* promoter hypomethylation (23.2% of patients) had significantly shorter OS than those without hypomethylation [78]. The latter being an independent prognostic risk factor for low-risk MDS patients [78]. Interestingly, hypomethylation of *DNMT3A* resulted in shorter OS and this was confirmed to be an independent prognostic factor in MDS [111].

Patients with high methylation levels of a 10-gene signature displayed shorter OS rates and shorter progression-free survival (PFS) rates [112]. This prognostic model was confirmed independent of age, gender or IPSS score [112]. Significantly high promoter methylation at specific genes such as *SOX7* [70], *GPX3* [74], *miR-124* [91], *SOCS1* [113], *DLX4* [73], *DLX5* [114], *sFRP1/4/5* [77], *p73* [115], *VTRNA1-3* [116], *CDKN2B* [76,113], *HIC1, CDH-1, ER* [117], and *ABAT* [69] is associated with low OS rates and/or poor prognosis in MDS patients, and many are independent prognostic factors for MDS. Some of these genes (*SOX7, GPX3, miR-124-1/-2,* and *CDKN2B*) also show higher levels of methylation as MDS progresses to secondary AML.

DNA methylation levels also correlate with MDS prognostic risk groups. High methylation index which examines global methylation levels in promoters and gene bodies, was significantly increased in higher-risk IPSS-R MDS patients [118]. *FOXO3* and *CHEK2* promoter methylation were also associated with high-risk parameters, with no methylation in these sites in healthy controls [119]. Moreover, *SHP-1* [120], *DLC-1* [75], *HRK* [121] and *SOX17* [72] promoter hypermethylation have also been shown to associate with high-risk MDS. Methylation at a specific site in the genome has also been linked to a better prognosis in MDS. Hypermethylation in a region preceding the *MEG3* gene before the commencement of AZA therapy in 50% of MDS patients was associated with longer PFS [79]. Therefore, DNA methylation changes in regulatory regions of specific genes may hold promise in predicting patient prognosis in MDS.

### 7.2. miRNAs That Predict Prognosis

The associations found between miRNA expression and patient risk groups, progression, and survival at different stages of disease progression have also been described in MDS (Table 2). The expression of a 10-miRNA signature and the expression of miR-15a and miR-16 have been shown to closely associate with prognosis scoring, permitting discrimination between lower and higher risk MDS cases [81,84]. Increased expression of miR-181 family members was also observed in higher risk MDS patients, and this overlapped with AML [81]. Moreover, the expression of 5 miRNAs, including three members of the miR-181 family, was able to identify MDS patients at higher risk of progression [122]. Differences in the expression of miRNAs between risk groups has also been observed. Higher-risk MDS patients displayed decreased expression of miR-17-5p and miR-20a compared to low-risk patients and let-7a was under expressed in patients with intermediate or high-risk MDS [123]. Lower expression of miR-21, miR-126, miR-146b-5p and miR-155 was found in IPSS low-intermediate risk MDS compared to higher-risk patients [93]. In addition, the circulating levels of miR-27a-3p, miR-150-5p, miR-199a-5p, miR-223-3p and miR-451a were decreased in higher-risk MDS and this was linked to prognosis.

High expression of miR-126, miR-155 and miR-124a [93], miR-661 [89], miR-100-5p [124], miR-194-5p and miR-320a [85], miR-181 family [81,84], miR-125a and miR-99b [125], and miR-22 [126] are linked to poor survival in MDS. Both miR-125a and miR-99b have been shown to activate NF-κB in vitro [125], and TET2 is a known target of miR-22 [126]. In contrast, low expression of miR-194-5p [85] and miR-126* (also known as miR-126-5p and originally named miR-123) [127] are associated with poor OS in MDS. Low expression of miR-126* was also linked to higher relapse rates and shorter PFS [127]. The multivariate analysis showed that miR-126*, age and the IPSS-R risk independently predicted PFS and OS [127]. Another group found that high expression of miR-17-5p and miR-20a predict good prognosis in MDS, as they were associated with increased OS of MDS patients [123].

The expression of circulating miRNAs has also been linked to PFS and OS in MDS. Recently, a small ncRNA signature in EVs containing low levels of miR-1237-3p and high levels of miR-548av-5p was associated with improved OS in MDS [83]. Moreover, lower expression of let-7a and miR-16 was significantly associated with PFS and OS [129]. However, only let-7a was a strong independent predictor of OS [129]. A 7-miRNA signature is also an independent predictor of survival in MDS with 75% accuracy and performs better than traditional risk models [130]. More recently, miR-451a expression was shown to be an independent predictor of PFS, and miR-223-3p expression led to significantly better OS [128].

### 7.3. lncRNAs That Predict Prognosis

There are very few reports investigating the link between expression of lncRNAs and prognosis in MDS to date (Table 2). Overexpression of MEG3 lncRNA was associated with poor prognosis in 50% of MDS cases, and after AZA therapy, MEG3 expression levels decreased and were closer to that of healthy controls [79]. Moreover, AML and MDS patients with higher HOXB-AS3 expression displayed significantly shorter OS [131]. In MDS patients this equated to adverse prognosis with median OS of 14.6 months with high HOXB-AS3 expression compared to 42.4 months [131]. Subgroup analysis showed that high HOXB-AS3 expression could only predict poor prognosis in the lower-risk MDS group [131]. High serum expression of the lncRNA KCNQ1OT1 [132] and high expression of LOC101928834 [101] have also been shown to associate with poor survival in MDS. Lastly, MDS patients with a modelled high lncRNA score displayed shorter OS and were more likely to progress to leukemia [99]. Therefore, increased expression of lncRNAs appears to negatively influence patient prognosis in MDS.

## 8. Epigenetic Modifiers as Biomarkers for Response to HMAs in MDS

HMAs such as DAC and AZA are used for the treatment of high-risk MDS patients. Although the use of HMAs has tripled survival rates for MDS patients, less than 50% of patients respond. Therefore, biomarkers that can accurately predict response to HMAs are important. Given the role that DNA methylation, miRNA and lncRNA play in MDS pathogenesis, these are potential candidates.

### 8.1. DNA Methylation as a Biomarker for Treatment Response in MDS

MDS patients with mutations in epigenetic machinery such as *DNMT3A, TET2, IDH1* and *IDH2*, tend to respond well to HMA therapy [133,134]. These mutations tend to occur with other mutations, and typically remain stable during treatment with AZA, irrespective of treatment response [135]. MDS patients with a *TET2* mutation appear to respond better to HMAs, particularly if they do not have *ASXL1* clonal mutations [136,137]. In terms of global DNA methylation levels, the decrease in methylation globally during HMA therapy, as opposed to baseline levels, has been shown to predict better response to HMAs [112]. In contrast, another study observed stable global methylation levels as assessed by LINE-1 methylation before and after AZA treatment in MDS patients who responded to AZA [138]. This observation could be due to differences in method of DNA methylation analysis, length of treatment and patient cohort.

Although global DNA methylation levels do not always appear to predict response to therapy, DNA methylation levels at specific genomic sites have been linked to treatment response (Table 3). A significant reduction in CpG methylation of *EZH2* (promoter) and *NOTCH1* (intragenic) was shown at best haematologic response in MDS patients who responded to AZA [138]. Therefore, hypermethylation at these sites before treatment and subsequent hypomethylation during treatment may predict response to AZA therapy. High methylation and hence low expression of cytidine deaminase (*CDA*; detoxification of AZA) [139] *PLCB1* (cell signalling transduction) [140,141,142,143] or *CDKN2B* (cell cycle regulator) [144] before treatment, coupled with decreased methylation and increased gene expression following AZA, may predict a better clinical response / hematologic response, respectively. However, another study found that lower baseline levels of *CDKN2B* methylation occurred in AZA responders, and although AZA reduced methylation, this did not correlate with treatment response [145]. Methylation *BCL2L10* (apoptotic regulator) [146] may also predict response to HMAs, however its predictability is unclear. More recently, the reduction of *DLC-1* (Rho GTPase activator) methylation following AZA treatment was also associated with a better response to AZA in MDS patients [147].

Increased accumulation of the deoxyribonucleoside form of AZA (5-AZA-CdR) in DNA [148] and less incorporation of AZA into RNA [149] have been associated with better treatment response. Some of the non-responders to AZA failed to incorporate adequate levels of 5-AZA-CdR into DNA, whereas others had incorporation and DNA hypomethylation, but this resulted in no clinical benefit [148]. Therefore, it appears that response may not be exclusively due to incorporation into DNA and the extent of DNA demethylation, but also to the regions of the genome that have undergone demethylation. Moreover, no significant differences in methylation (promoter and gene body) were observed before AZA treatment in MDS patients, regardless of subsequent treatment response [118]. Sequential assessment of whole blood DNA methylation levels in MDS patients treated with AZA found that AZA responders showed significantly higher recovery of hypomethylated DNA at the time of next course of AZA compared to non-responders, who did not display normalised methylation levels [118].

In summary, methylation studies have shown global and gene-specific promoter hypermethylation in MDS (Table 3), but there seems to be conflicting evidence regarding the degree of global demethylation following hypomethylating treatment and hematologic response. Research is starting to focus on assessing methylation changes in not just promoter regions but also other genomic regions (gene bodies, intergenic and enhancer regions). Therefore, it appears that DNA methylation changes at several specific genomic sites may provide benefit in predicting response to HMAs in MDS patients in the future.

### 8.2. ncRNAs as Biomarkers That Predict Response to HMA Therapy in MDS

The expression of miRNAs in serum or blasts extracted from MDS patients may also be useful biomarkers for predicting response to HMA therapy (Table 3). Serum levels of miR-21 have been shown to predict response to HMAs (ROC 0.648), with low baseline expression observed in responders, and this was associated with improved overall response rate (ORR) and PFS [150]. Decreased expression of miR-100-5p and miR-133b and increased expression of miR-17-3p have also been found to predict better ORR [124]. Moreover, a plasma miRNA signature (miR-423-5p, miR-126-3p, miR-151a-3p, miR-125a-5p, miR-199a-3p) was recently shown to predict response to AZA [83].

In contrast, MDS patients with low expression of miRNAs that regulate DNMT1, such as miR-126*, displayed significantly lower response rates, higher relapse rates, and shorter PFS and OS [127]. Decreased expression of miR-126* over time was also associated with increased risk of secondary resistance to AZA [127]. Therefore, the expression of specific miRNAs at diagnosis may aid in stratifying patients into treatment groups, and miRNA profiling throughout treatment may also predict response and resistance to HMAs.

Similar to miRNAs, lncRNAs have the potential to be used as biomarkers for treatment response in MDS (Table 3). However, there is only one study to date that has found lncRNAs that are associated with response to HMAs. Increased expression of lncRNAs PU.1 and JPD2 led to a favourable clinical response to AZA [151]. More studies are needed that focus on ncRNAs to determine those that may help predict response to HMAs and patient outcomes such as PFS and OS.

## 9. Epigenetic Modifiers in the MDS Bone Marrow Micro-Environment (BMME)

The bone marrow microenvironment (BMME) consists of an array of cell types such as mesenchymal stromal cells (MSCs), bone progenitor cells, endothelial cells, neurons and immune cells [152,153]. Many of these cell types play a supportive role in normal haematopoiesis and show abnormal function in disease states such as MDS [154,155,156,157]. While most of the research in MDS has focused on myeloid blasts, other cell types in the BMME may not only be dysfunctional in MDS but may also be targets for novel therapies and/or contain biomarkers for diagnosis, prognosis, and predictors of response to therapy. The BMME has started to gain more attention recently in terms of its role in the pathogenesis of MDS. *DICER1* gene deletion (inhibits DICER mediated miRNA processing) in bone marrow osteoprogenitor cells in mice induced MDS and AML-like haematological characteristics [23], highlighting the importance of the BMME and specifically the role of miRNAs in the BMME in MDS.

### 9.1. DNA Methylation in BMME

Widespread changes in MSCs in bone marrow from MDS patients that have been observed include chromosomal abnormalities [158,159], dysfunction [158,159], high levels of inflammatory cytokines [159,160] and aberrant DNA hypermethylation [161,162,163] compared to healthy controls, with hypermethylation occurring preferentially outside of CpG islands [162]. Following AZA treatment, MSCs from MDS (including high-risk patients) display significantly decreased DNA methylation, regardless of haematological response [161,162]. This is interesting because it shows that AZA can decrease methylation in cell types other than blasts, particularly MSCs that have a low proliferative rate. Moreover, only MSCs from MDS patients that reach complete remission seem to restore their normal phenotype and function compared to healthy donor MSCs [161]. MSCs that fail to respond to HMAs are associated with MDS patients with rapid and adverse disease progression [163]. Hypermethylation of *FRZB* has also been shown to decrease its expression in MDS stroma, leading to activation of WNT/β-catenin signalling in CD34+ cells from advanced cases of MDS and is associated with adverse prognosis (Figure 2) [162]. Methylation of *SPINT2/HAI-2* gene in stromal cells was shown to cause low expression, leading to enhanced adhesion and survival of CD34+ cells, potentially via interactions with specific integrins (Figure 2) [164]. Treatment with AZA increased *SPINT2/HAI-2* gene expression in MDS stromal cells but not stromal cells from healthy donors (Figure 2) [164]. Therefore, DNA methylation in stromal cells plays an important role in the cross talk between MDS blasts and their BMME, influencing cancer cell survival and progression.

Methylation levels of the *PD-1* promoter in CD8+ T-cells have been shown to influence response to HMAs. Demethylation of *PD-1* and subsequent PD-1 expression was observed in peripheral blood T-cells during AZA treatment (Figure 2) [165]. This significantly correlated with worse ORR and a trend towards shorter OS. In addition, patients that did not respond to AZA displayed significantly higher baseline *PD-1* methylation levels compared to healthy controls [165]. Therefore, HMAs influence PD-1 expression in T-cells and associated immune response against MDS blasts. Thus, these patients may benefit from a PD-1 pathway inhibitor to help reactivate the immune system.

### 9.2. miRNA and lncRNA in BMME

There are limited studies examining the role of miRNAs and lncRNAs in MDS-MSCs in diagnosis, prognosis, and treatment response. This field is still in its infancy. However, there are some reports of differential expression in MDS-MSCs and altered functions. Global downregulation of miRNA expression was observed in MDS-MSCs from patients compared to healthy controls [166]. Three miRNAs (miR-155, miR-181a and miR-222) had significantly decreased expression in MDS-MSCs compared to healthy donors and these are known to target DICER1 and DROSHA, members of the canonical miRNA biogenesis pathway [166]. Interestingly, DICER1 and DROSHA expression were decreased in MDS-MSCs [166]. Therefore, changes in miRNA expression in MSCs may influence hematopoietic cell functions as these cell types interact directly and via microvesicles. MSCs from MDS patients have also shown impaired proliferation, differentiation and differential miRNA expression compared to healthy controls [168]. DICER1, miR-30d-5p, miR-222-3p and miR-30a-3p displayed significantly decreased expression, and miR-4462 was overexpressed in MDS-MSCs [168].

Exosomes and microvesicles are involved in intercellular communication via release of their cargo once cell uptake has occurred [168]. MSCs from MDS patients showed overexpression of miR-10a and miR-15a within their exosomes, and these miRNAs were incorporated into CD34+ cells, modifying the expression of MDM2 and p53, leading to increased CD34+ cell viability and clonogenic capacity (Figure 2) [168]. Therefore, exosomes containing miRNAs released from MDS-MSCs are capable of being incorporated into hematopoietic progenitor cells and influence cellular functions. This provides another mechanism of cross talk between BMME and MDS blasts / progenitor cells and suggests that BMME may be a useful source of markers for diagnosis and prognosis, as well as provide novel therapeutic targets for MDS.

## 10. Epigenetic Modifiers in MDS: Conclusions and Future Directions

There are several DNA methylation, miRNA and lncRNA changes in MDS that may provide benefit in the diagnosis, prognosis and selection of therapies for MDS patients. These markers may also serve as therapeutic targets, leading to the development of novel targeted therapies, and may also provide benefit as markers for response to new targeted therapeutics currently being tested in the clinic such as BCL2 inhibitors and checkpoint inhibitors. Although the markers mentioned in this review show promise as biomarkers for MDS, their applicability in the clinic still warrants further investigation. More importance should be placed on studies with data from large and multiple patient cohorts, use of non-invasive methods (serum/serum EVs) and those that have displayed a high level of sensitivity and specificity as a biomarker in MDS.

Ideally, a clinical test would consist of panels of serum markers—multiple miRNA and lncRNA markers on one panel to assess expression in peripheral blasts and/or EVs, and another panel assessing multiple DNA methylation markers on DNA extracted from peripheral blasts. This would allow non-invasive testing using blood samples for diagnosis, prognosis, and tracking of treatment response. From a simple blood sample, DNA and RNA from peripheral blasts and RNA from peripheral blasts and/or EVs could be extracted and applied to next-generation sequencing (targeted amplicon bisulphite sequencing for DNA) or real-time PCR targeted (bisulphite PCR for DNA methylation) panels, technologies that are already routinely used in clinical testing of MDS patient samples.

Finding robust and reproducible markers for diagnostics and prognostics will ultimately improve clinical management and appropriate use of resources. Large prospective cohort studies will be needed to establish epigenetic modifiers as clinically useful biomarkers. With the advent of single-cell technologies and multi-omics over the last decade there is now the opportunity to not only delve deeper into how epigenetic processes collectively contribute to MDS pathogenesis but also examine the heterogeneity that exists within different cell types in a single patient [169,170,171,172]. This would also involve investigating epigenetic changes in the BMME (MSCs, T-cells) and peripheral blood (exosomes), instead of mainly focusing on MDS blasts within the bone marrow. We envision for the future clinically relevant epigenetic signatures capable of aiding diagnosis, prognosis and predictive of treatment responses in MDS, ultimately improving the quality of life and survival of MDS patients.

## Figures and Tables

**Figure 1 cancers-12-03128-f001:**
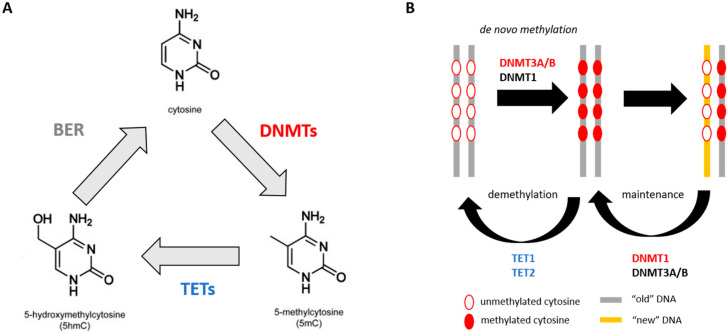
General schematic of DNA methylation and demethylation in somatic cells. (**A**) A methyl group is added to the 5′ carbon of a cytosine ring by DNMT enzymes giving rise to 5mC. This methyl group is oxidised by TET enzymes resulting in 5hmC which undergoes further oxidisation and base-excision repair (BER) to convert back to an unmodified cytosine. (**B**) De novo methylation is predominantly carried out by DNMT3A/B enzymes and during DNA replication methylation marks present on the template strand (“old” DNA) are copied onto the daughter strand (“new” DNA) mainly by DNMT1 enzyme. TET1 and TET2 enzymes instigate DNA demethylation via oxidisation.

**Figure 2 cancers-12-03128-f002:**
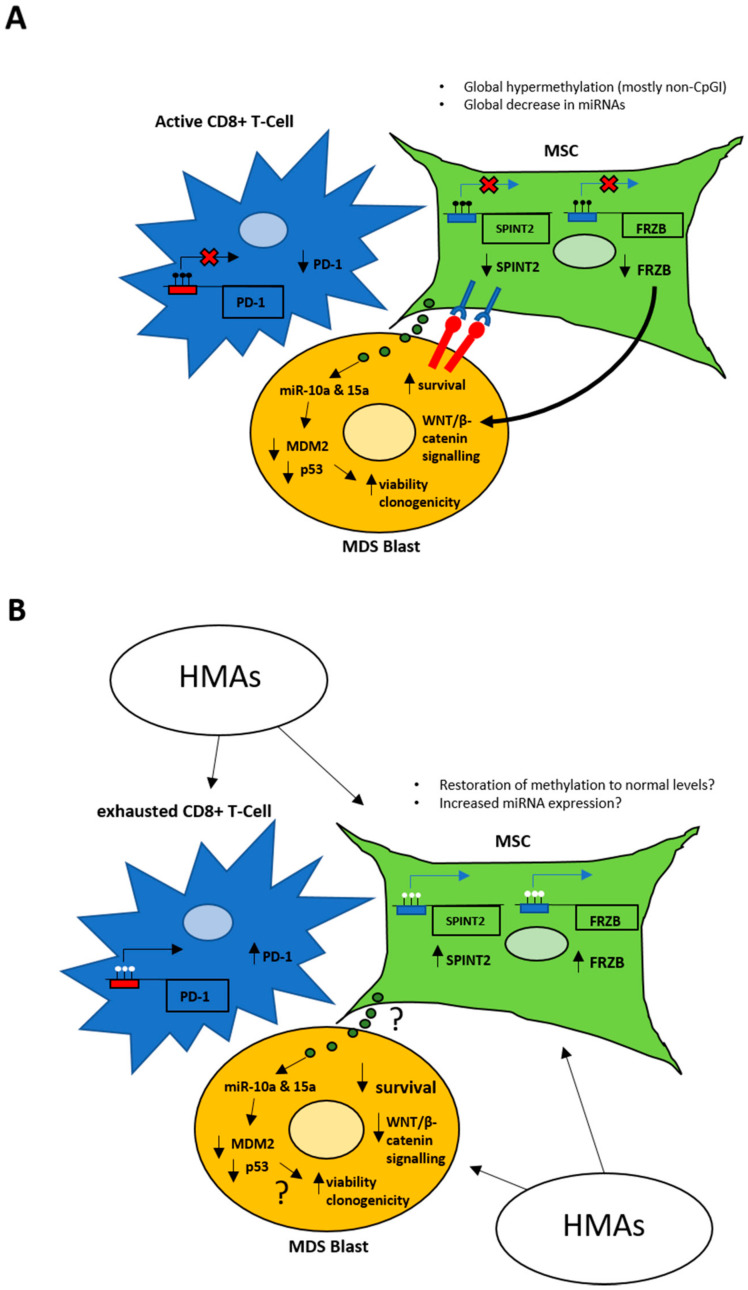
Schematic depicting interactions between MDS blasts and their bone marrow micro-environment. (**A**) In MDS, mesenchymal stromal cells (MSCs; green) display global hypermethylation and a global decrease in miRNA expression compared to healthy controls. They also have hypermethylation of *SPINT2*, resulting in decreased SPINT2 expression leading to adhesion to MDS blasts (yellow) via integrins and increased blast cell survival. Hypermethylation of *FRZB* is also observed leading to decreased FRZB expression in MSCs which increases WNT/beta-catenin signalling in MDS blasts. MSCs also release exosomes containing miR-10a and miR-15a, resulting in decreased MDM2 and p53 expression in MDS blasts and enhanced viability and clonogenicity. CD8+ T-cells (blue) in the area generally show hypermethylation at the *PD-1* promoter, resulting in decreased PD-1 expression and activated T-cells. (**B**) Treatment with HMAs such as decitabine (DAC) or AZA may result in restoration of methylation levels back to that of healthy controls and potentially lead to increased miRNA expression. HMA treatment results in demethylation of *SPINT2* and *FRZB* promoters, allowing expression of SPINT2 and FRZB and therefore disrupting some interactions between MSCs and MDS blasts, leading to decreased survival of blasts. The effect of HMAs on exosome release from MSCs and subsequent effects on MDS blast function are unknown. Treatment with HMAs also cause demethylation of *PD-1* promoters in CD8+ T-cells resulting in increased PD-1 expression and inhibition of T-cell activity. Adapted using information from [161,162,163,164,165,166,167].

**Table 1 cancers-12-03128-t001:** Epigenetic modifiers that may aid in the diagnosis of myelodysplastic syndrome (MDS).

Epigenetic Modifier	Diagnosis
*DNA Methylation*	High global levels of methylation (5mC) in bone marrow cells in 85% of MDS cases [65]
	Hypermethylation at enhancers, particularly in patients with *TET2* loss of function mutations [66,67]
	Increased CpG promoter methylation: *ABAT* [69], *DAPP1, FADD, LRRFIP1, PLBD1, SMPD3* [68] *SOX7* (55% of MDS patients, absent in controls) [70] *SOX17* (58% patients) [72] *ID4* [71] *DLX4* [73] *GPX3* [74] WNT antagonists (*sFRP1, sFRP2, sFRP4, sFRP5, DKK-1 and DKK-3*) [77] *CDKN2A* and *CDKN2B* [76] *DLC-1* (55% of patients) [75]
	Hypomethylation of *let-7a-3* promoter in MDS vs. controls [78]
*miRNA*	Overexpression of DLK1-DIO3 region (large miRNA cluster and lncRNA MEG3 promoter) in 50% of high-risk MDS who progressed to AML with myelodysplasia [79]
	miRNA signature that discriminated MDS from healthy controls: miR-378, miR-632, miR-636 [80], let-7 family [81]
	Higher percentage of miRNAs expressed in low-risk MDS compared to controls or high-risk MDS [82]
	Increased expression of miR-34a, miR-125a and miR-150 and miRNAs clustered on 14q32 in MDS [83]
	Increased expression in MDS: miR-17-92 cluster [84], miR-222, miR-10a [81], miR-194-5p, miR-320a [85], miR-21 [86,87], miR-34b [88], miR-661 [89], miR-720 [87], miR-205-5p [90]
	Decreased expression: miR-124 [91], miR-146a, miR-150, let-7e [81], miR-143 [92], miR-671-5p, miR-BART13 [87], miR-155, miR-182, miR-124a, miR-200c, miR-342-5p and let-7a [93]
	EV cargo from transfusion-dependent MDS cases – higher numbers of small RNAs and miRNAs, upregulated expression: miR-584J, miR-4485; down-regulation of: miR-28, let-7d [94]
	21 exosomal miRNA signature strongly associated with MDS [95]
*lncRNA*	lncRNA (lncENST00000444102) and ABAT were significantly downregulated in MDS [96]
	Significantly decreased LEF1-AS1 in MDS [97]
	linc-BDH1-1, linc-FAM75A7-7, linc-HHLA2-2, linc-JMJD1C-3, linc-PRKD1-2 and linc-RPIA aberrantly expressed in MDS [98]
	High expression of TC07000551.hg.1, TC08000489.hg.1, TC02004770.hg.1, and TC03000701 in MDS [99]
	Overexpression of CCAT2 [100]
	Novel lncRNA LOC101928834 upregulated in MDS bone marrow cells, could discriminate MDS-RAEB patients from controls (AUC 0.9048) [101]
	590 downregulated lncRNAs and 372 upregulated lncRNAs in MDS; co-ordinated and abnormal lncRNA and mRNA transcriptomes [102]

Extracellular vesicle (EV).

**Table 2 cancers-12-03128-t002:** Epigenetic modifiers associated with MDS patient prognosis.

Epigenetic Modifier	Prognosis
*DNA Methylation*	High global DNA methylation levels → decreased OS and increased progression to AML [108,109]
	MDS patients grouped based on DNA methylation profiles correlates with OS [110]
	Hypomethylation of *CD93* → shorter OS [110]
	Hypermethylation of a 10 gene signature (*CDH1, CH13, ER-alpha, NOR1, NPM2, OLIG2, p15INK4B, PGRA, PGRB and RIL*) → shorter OS and PFS [112]
	*SOX7* [70], *GPX3* [74], *SOCS1*, *CDKN2B* [113], *miR-124* [91], *DLX4* [73], *DLX5* [114], *sFRP1/4/5* [77], *p73* [115], *VTRNA1-3* [116] and *ABAT* [69] promoter methylation → decreased OS and increased progression to AML
	Hypermethylation of *p15INK4B, HIC1, CDH1,* and *ER* → poor prognosis [117]
	Hypermethylation of *CDKN2B* associated with disease progression and leukemic transformation [76]
	High methylation index (promoter and gene body methylation) → very high risk MDS [118]
	*SHP-1* promoter hypermethylation → high-risk MDS [120]
	*FOXO3* and *CHEK2* promoter methylation → high risk indicators [119]
	Increased methylation of *DLC-1* in high-risk MDS vs. low-risk MDS [75]
	Methylation of *HRK* [121] and *SOX17* [72] → advanced stage and high-risk MDS
	*MEG3* hypermethylation (in 50% of patients) → longer PFS [79]
	Hypomethylation of *DNMT3A* (57% of patients) [111] and *let-7a-3* (23.2% of patients) [78] → shorter OS
*miRNA*	10 miRNA signature (miR-181a/b/c/d, miR-221, miR-376b, miR-125b, miR-155, miR-130a and miR-486-5p) discriminated between low and high-risk MDS [81]
	High miR-15a (BM) and low miR-16 (PB) in high-risk MDS [84]
	Increased miR-181 family expression → Higher-risk MDS and progression to AML [81]
	5 miRNAs (miR-4865p, miR-181a-5p, miR-181b-5p, miR-199b-5p, miR-181d-5p) predicted progression to AML [122]
	Decreased expression of miR-17-5p and miR-20a, let-7a → High-risk MDS [123]
	Lower expression of miR-21, miR-126, miR-146b-5p, miR-155 in low-risk vs. high-risk MDS [93]
	High expression of miR-126, miR-155 and miR-124a [93], miR-661 [89], miR-100-5p [124], miR-194-5p, miR-320a [85], miR-181 family [81,84], miR-125a, miR-99b [125] and miR-22 [126] → poor survival
	Low expression of miR-194-5p → poor OS [85]
	Low expression of miR-126* → shorter OS, PFS and increased relapse [127]
	High expression of miR-17-5p and miR-20a → increased OS / good prognosis [123]
	Decreased circulating levels of miR-27a-3p, miR-150-5p, miR-199a-5p, miR-223-3p and miR-451a → High risk MDS and poor prognosis [128]
	High plasma miR-451a → independent predictor of longer PFSHigh plasma miR-223-3p levels significantly associated with better OS [128]
	High circulating Let-7a and miR-16 levels → poor PFS and OS [129] Plasma 7 miRNA signature (high let-7a, miR-144, miR-16, miR-25, miR-451 and low miR-651, miR-655; 75% accuracy) → poor survival [130]
*lncRNA*	Overexpression if MEG3 (50% of patients) → poor prognosis [79]
	Overexpression of HOXB-AS3 → shorter OS, poor prognosis for low-risk MDS [131]
	High LOC101928834 expression [101] and high serum KCNQ1OT1 expression [132] → poor survival
	High lncRNA score (TC07000551.hg.1, TC08000489.hg.1, TC02004770.hg.1, and TC03000701.hg.1) → poor OS and likely to progress to leukemia [99]

Overall survival (OS); progression-free survival (PFS); bone marrow (BM); peripheral blood (PB).

**Table 3 cancers-12-03128-t003:** Epigenetic modifiers that can predict response to hypomethylating agents (HMAs) in MDS.

Epigenetic Modifier	Treatment Response
*DNA Methylation*	*DNMT3A, TET2* [136,137], *IDH1, IDH2* mutations → better response to HMAs [133,134]
	Loss of methylation during treatment → better response to HMAs [112]
	AZA responders showed stable global methylation levels before and after treatment [138]
	Hypermethylation of *EZH2* (promoter) and *NOTCH1* (intragenic) before treatment and hypomethylation after treatment → best cytological response to AZA [138]
	Hypermethylation of *CDA* [139], *CDKN2B* [144] or *PLCB1* [140,141,142,143] before treatment → better response to HMAs
	Reduced *DLC-1* methylation after AZA → better response to AZA [147]
	Better recovery of methylation at time of next course of AZA → better response to AZA [118]
	Increased 5-AZA-CdR incorporation into DNA [148] and less AZA incorporation into RNA [149] → better response to AZA
*miRNA*	Low serum expression of miR-21 before treatment → better response to HMAs and PFS [150]
	Decreased expression of miR-100-5p and miR-133b, and increased miR-17-3p in MDS BM cells → predict better ORR [124]
	Plasma miRNA signature (miR-423-5p, miR-126-3p, miR-151a-3p, miR-125a-5p, miR-199a-3p) → predict response to AZA [83]
	Low expression of miR-126* → lower response rate, higher relapse rate, shorter PFS and OS [127]
*lncRNA*	Upregulation of PU.1 an JPD2 expression → better clinical response to AZA [151]

Hypomethylating agent (HMA); overall survival (OS); overall response rate (ORR); progression-free survival (PFS); azacytidine (AZA).

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
