# Peer review of "Unravelling the Epigenome of Myelodysplastic Syndrome: Diagnosis, Prognosis, and Response to Therapy"

_cancers, 2020, doi:10.3390/cancers12113128_

Round 1
Reviewer 1 Report
The manuscript "Unravelling the epigenome of Myelodysplastic Syndrome: diagnosis, prognosis, and response to therapy" is an interesting work that needs several revisions in order to be really useful in the field. Generally, the literature should be updated, the images should be improved and the characteristics of the patients should be better explained. All the specific revisions are indicated below:
Page 2: line 44. Change in: MDS [3]. However,
Page 2: line 47: This review focus on... it is repeated more than once (see also page 3, line 111 and page 4, lines 176.177): how many focuses does the review have?
Page 2: line 57: genetic and morphological for diagnosis: there is a space between and... and morphological to be deleted and a missing word after morphological
Page 2: lines 65-76: references 9-12 are very old and new data emerged. The entire paragraph has to be updated and reprhrased
Page 2: line 75: Change in: transplant. However,
Page 2: line 86: TP53: brand new data about TP53 mutations just appeared in nature Medicine
Page 3: lines 93-98 can be moved as the beginning of the following paragraph
Page 3: line 111: see comment on page 2, line 44
Page 3: lines 114-125: please add a schematic figure
Page 3: line 133: delete space between influence and global
Page 4: line 157: For e.g. is redundant. Please choose between "For example" and "e.g."
Page 4: line 171: For e.g. is redundant. Please choose between "For example" and "e.g."
Page 4: lines 176-177: see comment on page 2, line 44
Page 4: lines 176-177: the sentence is questionable, now MDS patients are diagnosed also with early disease. Please explain better or rephrase with up-to-date references
Page 5: lines 219-220: correct in: myelodysplasia-related
Page 6: line 236: replace & with and. If it is a cluster, please define it better
Page 6: line 243: please add reference
Page 6, line 245: change in "EVs, such as exosomes, contain
Page 6, line 245: explain "cargo"
Page 6, line 249: miR-584J is the gene name, if it is referred to the mRNA it should be miR-584j. Please check and correct if appropriate
Page 6, line 259: please explain the shift better and why this would be important
Page 6, line 266: delete space between linc-RPIA and [92]
Page 7, line 284: which kind of MDS patients? All of them? Low? High? Please define them better
Page 7, line 286-287: change in: AML. However,
Page 8, line 305: change high risk in higher risk
Page 8, line 316: change in "10-miRNA signature"
Page 8, line 322 and 325: change high risk in higher risk
Page 8, line 328: replace & with and. If it is a cluster, please define it better
Page 8, line 332: please add: miR-126* (also known as miR-126-5p and originally named miR-123) for the readers unfamiliar with name changes
Page 8, line 340: change in: [118]. However,
Page 8, lines 337-342: are the miRNAs associated with signature up- or down- regulated?
Page 10, line 360: Decitabine and Azacytidine have already been defined (Page 2, line 72). Here use the abbreviation. Moreover, decide one of the names for AZA and use that one throughout all the text
Page 11, line 371: change in "HMA therapy, as opposed to"
Page 11, line 382: reference 129 is inappropriate. The paper delas with a genetic monoallelic deletion and not to promoter methylation. Several references could apply: Follo My et al. PNAS 2009, Fili et al. Clin Cancer Res 2013, Follo MY et al. Leukemia 2012, Cocco L et al. J Leuk Biol 2015. The authors should refer them instead
Page 11, line 384: please specify the "better response": which kind of response? OS, PFS, LFS?
Page 11, line 394: change in "but also to the regions"
Page 11, line 399: insert comma between "non-responders" and "who did"
Page 11, lines 401-406: the sentence is questionable. There are recent reports about the prognostic significance of specific genes and promoter regions, although other genetic regions are investigated. Autoros should comment on PLCB1, EZH2, DLX5, CDKN2B among the most recent ones, and many more as examples. As for global methylation, even in this case there are recent reports about its importance: 5-methylcitosine expression (Sucic M et al. Ann Clin Lab Sci 2019), DNA methylation(Tobiasdon M et al. Oncotarget 2017, Calvo X et al Leuk Res 2014)
Page 11, line 407: change ncRNAs in miRNAs, to uniform the title with the other paragraphs
Page 11, line 409: delete "as"
Page 11, line 412:, delete comma aftwr miR-133b
Page 13, line 441: add "in BMME" to uniform the title paragraph with the others in the text
Page 13, line 442: change in " MDS patients that have been observed include"
Page 13, line 446: change in "MSCs from MDS" and specify which kind of patients (HR? LR?)
Page 13, line 454: add comma between "expression" and "leading"
Page 13, line 467: add "in BMME" to uniform the title paragraph with the others in the text
Page 13, line 469: change in: infancy. However,
Page 13, line 481: change "releasing" into "release of"
Page 13, line 483: add comma after CD34+ cells
Page 14, line 487: delete "the" between "between" and "BMME"
Page 14, line 488: delete "the" between "that" and "BMME"
Page 14, line 488: change in "as well as provide novel"
Author Response
The manuscript "Unravelling the epigenome of Myelodysplastic Syndrome: diagnosis, prognosis, and response to therapy" is an interesting work that needs several revisions in order to be really useful in the field. Generally, the literature should be updated, the images should be improved and the characteristics of the patients should be better explained. All the specific revisions are indicated below:
We thank reviewer 1 for their thorough review of the manuscript and appreciate their comments and suggestions which we have addressed. We have updated the literature and improved / added images where suggested. We have better explained the patient characteristics as suggested throughout the manuscript. However, this has been difficult to explain in detail given that the aim of the review is to provide an overview of DNA methylation and non-coding RNA biomarkers in MDS generally. Where the expression of markers is important to a particular group of MDS patients, we have tried to make this clear in both the text and tables. Please see lines 247 and 326 as well as table 1, lines 314, lines 336-349 and table 2 for evidence of this. We also thank you for your specific comments that we have addressed below.
Page 2: line 44. Change in: MDS [3]. However,
Changed as suggested.
Page 2: line 47: This review focus on... it is repeated more than once (see also page 3, line 111 and page 4, lines 176.177): how many focuses does the review have?
Sentences outlining focus of the review on page 2 and 3 have been removed due to duplication.
Page 2: line 57: genetic and morphological for diagnosis: there is a space between and... and morphological to be deleted and a missing word after morphological
The space has been deleted and “abnormalities” added after “morphological.”
Page 2: lines 65-76: references 9-12 are very old and new data emerged. The entire paragraph has to be updated and reprhrased
The entire paragraph has been rephrased; however, some references have not been updated due to them being the original articles testing those therapeutics. We have also added in a sentence about emerging targeted therapies with references to early results from current clinical trials.
Lines 65-80: The treatment for low risk MDS consists of supportive care such as blood and platelet transfusions, antibiotics, erythropoietin (EPO) injections to stimulate red blood cell production and granulocyte colony-stimulating factor (G-CSF) to stimulate white blood cell production [8, 9]. Hypomethylating agents are used in higher risk patients to achieve remission, improve blood counts, reduce transfusion dependence, and bridge suitable patients to allogenic transplantation. Hypomethylating agents (HMAs) such as Azacytidine (AZA; Vidaza) [10] or Decitabine (DAC; Dacogen) are used in frontline therapy [11]. While HMAs are effective in around 50% of MDS patients in the short term, many patients become resistant to therapy and progress to bone marrow failure or to AML [12]. New targeted therapeutics such as BCL2 inhibitors (Venetoclax) [13-15] and immune checkpoint inhibitors [16-18] are currently being tested in combination with HMAs or as a monotherapy following HMA failure. In addition, Lenalidomide (Revlimid) is FDA approved for use in MDS with 5q deletion and immunosuppressive therapy or other chemotherapies (e.g. cytarabine) may be used as needed to improve normal blood cell production and reduce blast counts before stem cell transplantation, respectively [19]. The only curative treatment for MDS is stem cell transplant. However, most patients are too frail or have existing comorbidities which precludes them from undergoing this curative option [8, 10].
Page 2: line 75: Change in: transplant. However,
Change made as suggested.
Page 2: line 86: TP53: brand new data about TP53 mutations just appeared in nature Medicine
Thank you for suggesting this reference. I have made the following changes (lines 86-91):
These changes can also co-occur as complex karyotypes and are often accompanied by TP53 mutations [20, 27-29]. Interestingly, a recent study has highlighted an important role for bi-allelic TP53 mutations [29]. There was no difference in outcome for patients with wild-type or mono-allelic mutations of TP53. However, multiple mutations in TP53 were able to predict outcomes independent of the revised international prognostic scoring system.
References 28 (https://doi.org/10.1038/s41375-018-0351-2) and 29 (https://doi.org/10.1038/s41591-020-1008-z) added.
Page 3: lines 93-98 can be moved as the beginning of the following paragraph
Changes made as suggested.
Page 3: line 111: see comment on page 2, line 44
This sentence has been removed.
Page 3: lines 114-125: please add a schematic figure
Schematic figure (Figure 1) with figure legend and in-text references has been added.
Page 3: line 133: delete space between influence and global
Space between influence and global was deleted.
Page 4: line 157: For e.g. is redundant. Please choose between "For example" and "e.g."
Changed to “For example”.
Page 4: line 171: For e.g. is redundant. Please choose between "For example" and "e.g."
Changed to “For example”.
Page 4: lines 176-177: see comment on page 2, line 44
Sentence changed to: Given the importance of DNA methylation and ncRNAs in cancer biology, epigenetic modifiers in MDS, including changes in DNA methylation, miRNAs and lncRNAs, and how they may aid in MDS diagnosis (Table 1), prognosis (Table 2) and predicting response to treatment (Table 3) will be discussed below.
Page 4: lines 176-177: the sentence is questionable, now MDS patients are diagnosed also with early disease. Please explain better or rephrase with up-to-date references
This paragraph has been rephrased and changed to incorporate CHIP and CCUS:
Lines 194-203:
Diagnostic testing is usually initiated once patients have become symptomatic and cytopenia’s are prominent. Some of the most mutated genes in MDS are members of the DNA methylation machinery such as DNMT3A, TET2, IDH1 and IDH2 [61]. Mutations in DNMT3A and TET2 have been observed in clonal haematopoiesis and early in MDS [62, 63]. These mutations often lead to global changes in DNA methylation or pronounced changes at specific genomic sites. Mild cytopenia’s without overt features of myelodysplasia within the bone marrow are now increasingly recognised such as clonal cytopenia’s of uncertain significance (CCUS) [64]. Whether DNA methylation signatures may have the potential to aid in the recognition pre-MDS states such as CCUS or Clonal Haematopoiesis of Indeterminate Potential (CHIP) needs to be determined by prospective studies [64].
Page 5: lines 219-220: correct in: myelodysplasia-related
Added hyphen between “myelodysplasia” and “related.”
Page 6: line 236: replace & with and. If it is a cluster, please define it better
Replaced “&” with “and.” This is not a cluster – they are separate miRNAs from the same study.
Page 6: line 243: please add reference
Lines 260-262 - References added: However, only miR-21 [89, 90], miR-150 [83, 87] and let-7 miRNAs [83, 95] have been found to be differentially expressed in MDS by more than one research group.
Page 6, line 245: change in "EVs, such as exosomes, contain
Added a comma after EVs and after exosomes as suggested.
Page 6, line 245: explain "cargo"
Lines 264-265 - The sentence has been changed to: EVs, such as exosomes, contain cargo that consists of small RNAs and miRNAs that can be delivered to cells via intercellular communication [97].
Page 6, line 249: miR-584J is the gene name, if it is referred to the mRNA it should be miR-584j. Please check and correct if appropriate
Changed to miR-548j.
Page 6, line 259: please explain the shift better and why this would be important
Changed sentence to (lines 277-281): Since then, more studies have analysed the expression of specific lncRNAs and there has also been a shift towards exploring the global profile of lncRNA expression in MDS. Knowledge of global changes in lncRNAs is important to better understand how they are globally influencing cancer cell functions given their complexity in mode of action and potential to interact with multiple targets.
Page 6, line 266: delete space between linc-RPIA and [92]
Space deleted as suggested.
Page 7, line 284: which kind of MDS patients? All of them? Low? High? Please define them better
Changed sentence to: DNA methylation changes have also been associated with predicting prognosis in MDS patients of various subgroups, particularly with regards to overall survival (OS) (Table 2).
This is the first sentence of the paragraph which incorporates all MDS patients regardless of risk grouping. Throughout this section we state whether DNA methylation changes predict prognosis in particular risk groups or regardless of risk grouping where appropriate. For example, lines 320-322.
Page 7, line 286-287: change in: AML. However,
Changed to AML. However,
Page 8, line 305: change high risk in higher risk
Changed to “higher-risk”
Page 8, line 316: change in "10-miRNA signature"
Changed to 10-miRNA signature as suggested.
Page 8, line 322 and 325: change high risk in higher risk
Changed to “higher-risk”
Page 8, line 328: replace & with and. If it is a cluster, please define it better
Replaced “&” with “and”
Page 8, line 332: please add: miR-126* (also known as miR-126-5p and originally named miR-123) for the readers unfamiliar with name changes
Added (also known as miR-126-5p and originally named miR-123) as suggested after miR-126*.
Page 8, line 340: change in: [118]. However,
Changed as suggested.
Page 8, lines 337-342: are the miRNAs associated with signature up- or down- regulated?
Recently, a small non-coding RNA signature in EVs containing low levels of miR-1237-3p and high levels of miR-548av-5p was associated with improved OS in MDS [87].
Page 10, line 360: Decitabine and Azacytidine have already been defined (Page 2, line 72). Here use the abbreviation. Moreover, decide one of the names for AZA and use that one throughout all the text
The words Decitabine and Azacytidine were removed and their abbreviation used instead. 5-Azacytidine changed to Azacytidine in the abstract.
Page 11, line 371: change in "HMA therapy, as opposed to"
Comma added after HMA therapy, as suggested.
Page 11, line 382: reference 129 is inappropriate. The paper delas with a genetic monoallelic deletion and not to promoter methylation. Several references could apply: Follo My et al. PNAS 2009, Fili et al. Clin Cancer Res 2013, Follo MY et al. Leukemia 2012, Cocco L et al. J Leuk Biol 2015. The authors should refer them instead
We thank the reviewer for picking up on this mistake. We have removed the wrong reference and added all 4 suggested references in (now lines 404-405): High methylation and hence low expression of cytidine deaminase (CDA; detoxification of AZA) [139] PLCB1 (cell signalling transduction) [140-143]…
Page 11, line 384: please specify the "better response": which kind of response? OS, PFS, LFS?
Changed to: AZA, may predict a better clinical response or hematologic response, respectively.
Page 11, line 394: change in "but also to the regions"
Changed to “but also to the regions”
Page 11, line 399: insert comma between "non-responders" and "who did"
Added comma as suggested.
Page 11, lines 401-406: the sentence is questionable. There are recent reports about the prognostic significance of specific genes and promoter regions, although other genetic regions are investigated. Autoros should comment on PLCB1, EZH2, DLX5, CDKN2B among the most recent ones, and many more as examples. As for global methylation, even in this case there are recent reports about its importance: 5-methylcitosine expression (Sucic M et al. Ann Clin Lab Sci 2019), DNA methylation (Tobiasdon M et al. Oncotarget 2017, Calvo X et al Leuk Res 2014)
We acknowledge the reviewer’s concern and apologise for the lack of clarity. The purpose of this paragraph was to summarise results that were discussed in detail in the preceding section. We apologise that this was not clear, and have modified the text to avoid confusion:
This is a summary for the section and therefore has been changed to:
In summary, methylation studies have shown global and gene-specific promoter hypermethylation in MDS (Table 3), but there seems to be conflicting evidence regarding the degree of global demethylation following hypomethylating treatment and hematologic response. Research is starting to focus on assessing methylation changes in not just promoter regions but also other genomic regions (gene bodies, intergenic and enhancer regions). Therefore, it appears that DNA methylation changes at several specific genomic sites may provide benefit in predicting response to HMAs in MDS patients in the future.
We thank the reviewer for specific suggestions, and have included the references in our discussion, as follows:
PLCB1 – added to text and referenced Cocco et al 2015 and others as suggested (lines 404-405). High methylation and hence low expression of cytidine deaminase (CDA; detoxification of AZA) [139] PLCB1 (cell signalling transduction) [140-143]…
EZH2: Discussed at lines 400-402 - A significant reduction in CpG methylation of EZH2 (promoter) and NOTCH1 (intragenic) was shown at best haematologic response in MDS patients who responded to AZA [138].
DLX5: Hypermethylation in MDS, even higher with transition to sAML à linked to poor OS and poor leukemia-free survival (DOI: 10.1002/ctm2.29) – not directly linked to treatment response in MDS but have added to the DNA methylation prognostic section (lines 319-322: Significantly high promoter methylation at specific genes such as SOX7 [70], GPX3 [74], miR-124 [94], SOCS1 [113], DLX4 [73], DLX5 [114], sFRP1/4/5 [77], p73 [115], VTRNA1-3 [116], CDKN2B [76, 113], HIC1, CDH-1, ER [117], and ABAT [69] is associated with low OS rates and/or poor prognosis in MDS patients, and many are independent prognostic factors for MDS) and the prognosis table (Table 2).
CDKN2B: demethylation following treatment with AZA led to gene re-expression and hematologic improvement (doi: 10.3346/jkms.2011.26.2.207) – added to treatment response section. Lower baseline levels of CDKN2B methylation occurred in responders. AZA reduced methylation levels but this did not correlate with response (https://doi.org/10.1038/sj.leu.2404796) – added to section.
Lines 404-410: High methylation and hence low expression of cytidine deaminase (CDA; detoxification of AZA) [139] PLCB1 (cell signalling transduction) [140-143] or CDKN2B (cell cycle regulator) [144] before treatment, coupled with decreased methylation and increased gene expression following AZA, may predict a better clinical response / hematologic response, respectively. However, another study found that lower baseline levels of CDKN2B methylation occurred in AZA responders, and although AZA reduced methylation, this did not correlate with treatment response [145].
Page 11, line 407: change ncRNAs in miRNAs, to uniform the title with the other paragraphs
I have left this as ncRNAs as it incorporates both miRNAs and lncRNAs, both of which are part of this sub-section. Please see lines 156-160 for a definition of ncRNAs.
Page 11, line 409: delete "as"
Deleted “as”
Page 11, line 412: delete comma aftwr miR-133b
Deleted comma after miR-133b.
Page 13, line 441: add "in BMME" to uniform the title paragraph with the others in the text
Added “in BMME” to title.
Page 13, line 442: change in " MDS patients that have been observed include"
Changed to MDS patients that have been observed include… as suggested.
Page 13, line 446: change in "MSCs from MDS" and specify which kind of patients (HR? LR?)
Changed to: Following AZA treatment, MSCs from MDS (including high-risk patients) display significantly decreased DNA methylation, regardless of haematological response [161, 162].
Page 13, line 454: add comma between "expression" and "leading"
Comma added between “expression” and “leading”
Page 13, line 467: add "in BMME" to uniform the title paragraph with the others in the text
Added “in BMME” to title.
Page 13, line 469: change in: infancy. However,
Full stop added after infancy and comma added after However.
Page 13, line 481: change "releasing" into "release of"
Changed to release of.
Page 13, line 483: add comma after CD34+ cells
Comma added after “CD34+ cells”
Page 14, line 487: delete "the" between "between" and "BMME"
Made change as suggested.
Page 14, line 488: delete "the" between "that" and "BMME"
Made change as suggested.
Page 14, line 488: change in "as well as provide novel"
Changed to “as well as provide novel” as suggested.
Reviewer 2 Report
Bond et al present a comprehensive review of the emerging molecular pathogeneisis of myelodysplastic syndromes. It focusses on the evolving comprehension of epigenetic modifiers (in particular non-coding RNAs) and attempts to describe its relevance in terms of prognosis and treatment selection.
This article is a well-researched, concise and wide-ranging update in a complex field and provides the reader with a comprehensive overview of the emerging science, with a focus on providing bench to clinic relevance.
One area which could further be developed is a more comprehensive vision for real-world future diagnostic and prognostication strategies, ideally with a “weighted” stratification of the importance of the emerging data – this could be fleshed out further in the conclusion.
More on potential testing methods and a comment on their routine feasibility would also be useful. Are there 'shovel ready' technologies which could be implemented in routine clinical use?
These data provide scope for the development of new therapeutic targets which is not particularly mentioned in the article, and it may be worth adding a small section on this. In particular the relevance of treatments like BCL2 inhibitors or checkpoint inhibitors.
Having said that, the article is ultimately publishable in its current form.
Author Response
Bond et al present a comprehensive review of the emerging molecular pathogeneisis of myelodysplastic syndromes. It focusses on the evolving comprehension of epigenetic modifiers (in particular non-coding RNAs) and attempts to describe its relevance in terms of prognosis and treatment selection.
This article is a well-researched, concise and wide-ranging update in a complex field and provides the reader with a comprehensive overview of the emerging science, with a focus on providing bench to clinic relevance.
One area which could further be developed is a more comprehensive vision for real-world future diagnostic and prognostication strategies, ideally with a “weighted” stratification of the importance of the emerging data – this could be fleshed out further in the conclusion.
More on potential testing methods and a comment on their routine feasibility would also be useful. Are there 'shovel ready' technologies which could be implemented in routine clinical use?
The following was added (in red) to section 10 (conclusion) to give more of a comprehensive future view of diagnostic and prognostic testing with a weighting on the importance of the emerging data. A comment was also made with regards to routine feasibility and technologies that could be implemented / already are used in the clinic:
Lines 515-531: There are several DNA methylation, miRNA and lncRNA changes in MDS that may provide benefit in the diagnosis, prognosis and selection of therapies for MDS patients. These markers may also serve as therapeutic targets, leading to the development of novel targeted therapies, and may also provide benefit as markers for other targeted therapeutics currently being tested in the clinic such as BCL2 inhibitors and checkpoint inhibitors. Although the markers mentioned in this review show promise as biomarkers for MDS, their applicability in the clinic still warrants further investigation. More importance should be placed on studies with data from large and multiple patient cohorts, use of non-invasive methods (serum / serum EVs) and those that have displayed a high level of sensitivity and specificity as a biomarker in MDS.
Ideally, a clinical test would consist of panels of serum markers – multiple miRNA and lncRNA markers on one panel to assess expression in peripheral blasts and/or EVs, and another panel assessing multiple DNA methylation markers on DNA extracted from peripheral blasts. This would allow non-invasive testing using blood samples for diagnosis, prognosis, and tracking of treatment response. From a simple blood sample, DNA and RNA from peripheral blasts and RNA from peripheral blasts and/or EVs could be extracted and applied to next-generation sequencing (targeted amplicon bisulphite sequencing) or real-time PCR targeted (bisulphite PCR for DNA methylation) panels, technologies that are already routinely used in clinical testing of MDS patient samples.
These data provide scope for the development of new therapeutic targets which is not particularly mentioned in the article, and it may be worth adding a small section on this. In particular the relevance of treatments like BCL2 inhibitors or checkpoint inhibitors.
This has been covered in lines 522-526: There are several DNA methylation, miRNA and lncRNA changes in MDS that may provide benefit in the diagnosis, prognosis and selection of therapies for MDS patients. These markers may also serve as therapeutic targets, leading to the development of novel targeted therapies, and may also provide benefit as markers for response to new targeted therapeutics currently being tested in the clinic such as BCL2 inhibitors and checkpoint inhibitors.
BCL2 inhibitors and checkpoint inhibitors have also been added to the MDS treatment section (section 3 lines 70-72)
Having said that, the article is ultimately publishable in its current form.
Reviewer 3 Report
This review article dealing with epigenetics including non-coding RNA in myelodysplastic syndromes (MDS) is comprehensively written and well organized. I think that this is worth to be published in Cancers.
Some minor points should be added and revised especially for micro RNA.
Up and down of some micro RNAs are confusing. So please describe up/down of micro RNAs and their correlation with promoter methylation either in main text or table.
Please describe targets of micro RNAs in main text as well as figures if they are identified in MDS.
Author Response
This review article dealing with epigenetics including non-coding RNA in myelodysplastic syndromes (MDS) is comprehensively written and well organized. I think that this is worth to be published in Cancers.
Some minor points should be added and revised especially for micro RNA.
Up and down of some micro RNAs are confusing. So please describe up/down of micro RNAs and their correlation with promoter methylation either in main text or table.
We apologise for the confusion and have re-worded the paragraph to make this clearer and have stated where miRNA expression is correlated to promoter methylation if known. Please see below for changes:
Lines 251-259:
The following miRNAs have shown increased expression in MDS: miR-17-92 cluster [84], miR-222 & miR-10a [83], miR-194-5p (AUC 0.797) and miR-320a (AUC 0.729) [88], miR-21 [89, 90], miR-34b [91], miR-661 [92], miR-720 [90] and miR-205-5p (AUC 0.825) [93]. Conversely, downregulation of the following miRNAs has been observed in MDS: miR-124 [94], miR-155, miR-182, miR-124a, miR-200c, miR-342-5p and let-7a [95], miR-146a, miR-150 and let-7e [83], miR-143 [96], miR-671-5p and miR-BART13 [90]. In some cases, these changes in miRNA expression have been correlated to changes in DNA methylation in their promoters. Increased expression of miR-34b is associated with hypomethylation [91], and decreased expression of miR-124 is linked to increased DNA methylation [94].
Please describe targets of micro RNAs in main text as well as figures if they are identified in MDS.
Given the complexity of many mRNA targets of each miRNA and that this review is focussed on using miRNAs as markers, not their cellular functions, it’s beyond the scope of this review to list all known or potential miRNA targets. However, we have highlighted specific miRNA targets of particular interest in MDS within the text and figures where known. Please see Figure 2 and its legend (MDM2 and p53 are targets of miR-10a and miR-15a).
In text, please see lines 352-353: Both miR-125a and miR-99b have been shown to activate NF-κB in vitro [126], and TET2 is a known target of miR-22 [127].
Lines 498-500: Three miRNAs (miRNA-155, miRNA-181a and miRNA-222) had significantly decreased expression in MDS-MSCs compared to healthy donors and these are known to target DICER1 and DROSHA, members of the canonical miRNA biogenesis pathway [166].
Lines 508-511: MSCs from MDS patients showed overexpression of miR-10a and miR-15a within their exosomes, and these miRNAs were incorporated into CD34+ cells, modifying the expression of MDM2 and p53, leading to increased CD34+ cell viability and clonogenic capacity (Figure 2) [168].
Round 2
Reviewer 1 Report
The manuscript "Unravelling the epigenome of Myelodysplastic Syndrome: diagnosis, prognosis, and response to therapy" is now ready to be considered for publication.